# Markov Random Fields for Collaborative Filtering

**Harald Steck**
Netflix
Los Gatos, CA 95032
hsteck@netflix.com

## Abstract

In this paper, we model the dependencies among the items that are recommended to a user in a collaborative-filtering problem via a Gaussian Markov Random Field (MRF). We build upon Besag's auto-normal parameterization and pseudo-likelihood [7], which not only enables computationally efficient learning, but also connects the areas of MRFs and sparse inverse covariance estimation with autoencoders and neighborhood models, two successful approaches in collaborative filtering. We propose a novel approximation for learning sparse MRFs, where the trade-off between recommendation-accuracy and training-time can be controlled. At only a small fraction of the training-time compared to various baselines, including deep nonlinear models, the proposed approach achieved competitive ranking-accuracy on all three well-known data-sets used in our experiments, and notably a 20% gain in accuracy on the data-set with the largest number of items.

## 1 Introduction

Collaborative filtering has witnessed significant improvements in recent years, largely due to models based on low-dimensional embeddings, like weighted matrix factorization (e.g., [26, 39]) and deep learning [23, 22, 33, 47, 62, 58, 20, 11], including autoencoders [58, 33]. Also neighborhood-based approaches are competitive in certain regimes (e.g., [1, 53, 54]), despite being simple heuristics based on item-item (or user-user) similarity matrices (like cosine similarity). In this paper, we outline that Markov Random Fields (MRF) are closely related to autoencoders as well as to neighborhood-based approaches. We build on the enormous progress made in learning MRFs, in particular in sparse inverse covariance estimation (e.g., [36, 59, 15, 2, 60, 44, 45, 63, 55, 24, 25, 52, 56, 51]). Much of the literature on sparse inverse covariance estimation focuses on the regime where the number of data points $n$ is much smaller than the number of variables $m$ in the model ($n < m$).

This paper is concerned with a different regime, where the number $n$ of data-points (i.e., users) and the number $m$ of variables (i.e., items) are both large as well as $n > m$, which is typical for many collaborative filtering applications. We use an MRF as to model the dependencies (i.e., similarities) among the items that are recommended to a user, while a user corresponds to a sample drawn from the distribution of the MRF. In this regime ($n > m$), learning a sparse model may not lead to significant improvements in prediction accuracy (compared to a dense model). Instead, we exploit model-sparsity as to reduce the training-time considerably, as computational cost is a main concern when both $n$ and $m$ are large. To this end, we propose a novel approximation that enables one to trade prediction-accuracy for training-time. This trade-off subsumes the two extreme cases commonly considered in the literature, namely regressing each variable against its neighbors in the MRF and inverting the covariance matrix.

This paper is organized as follows. In the next section, we review Besag's auto-normal parameterization and pseudo-likelihood, and the resulting closed-form solution for the fully-connected MRF. We then state the key Corollary in Section 2.2.2, which is the basis of our novel sparse approximation outlined in Section 3. We discuss the connections to various related approaches in Section 4. The

empirical evaluation on three well-known data-sets in Section 5 demonstrates the high accuracy achieved by this approach, while requiring only a small faction of the training-time as well as of the number of parameters compared to the best competing model.

## 2 Pseudo-Likelihood

In this section, we lay the groundwork for the novel sparse approximation outlined in Section 3.

### 2.1 Model Parameterization

In this section, we assume that the graph $G$ of the MRF is given, and each node $i \in \mathcal{I}$ in $G$ corresponds to an item in the collaborative-filtering problem. The $m = |\mathcal{I}|$ nodes are associated with the (row) vector of random variables $X = (X_1, ..., X_m)$ that follows a zero-mean[1] multivariate Gaussian distribution $\mathcal{N}(0, \Sigma)$.[2] A user corresponds to a sample drawn from this distribution. As to make recommendations, we use the expectation $\mathbb{E}[X_i | X_{\mathcal{I} \setminus \{i\}} = x_{\mathcal{I} \setminus \{i\}}]$ as the predicted score for item $i$ given a user's observed interactions $x_{\mathcal{I} \setminus \{i\}}$ with all the *other* items $\mathcal{I} \setminus \{i\}$. When learning the MRF, maximizing the (L1/L2 norm regularized) likelihood of the MRF that is parameterized according to the Hammersley-Clifford theorem [18] can become computationally expensive, given that the number of items and users is often large in collaborative filtering applications. For this reason, we use Besag's auto-normal parameterization of the MRF [7, 8]: the *conditional* mean of each $X_i$ is parameterized in terms of a regression against the *remaining* nodes:

$$\mathbb{E}[X_i | X_{\mathcal{I} \setminus \{i\}} = x_{\mathcal{I} \setminus \{i\}}] = \sum_{j \in \mathcal{I} \setminus \{i\}} \beta_{j,i} x_j = x \cdot \mathbf{B}_{\cdot, i} \tag{1}$$

where $\beta_{j,i} = 0$ if the edge between nodes $i$ and $j$ is absent in $G$, i.e., the regression of each node $i$ involves only its neighbors in $G$. In the last equality in Eq. 1, we switched to matrix notation, where $\mathbf{B} \in \mathbb{R}^{m \times m}$ with $\mathbf{B}_{j,i} := \beta_{j,i}$ for $i \neq j$, and with a zero diagonal, $\mathbf{B}_{i,i} = 0$, as to exclude $X_i$ from the covariates in the regression regarding its own mean in Eq. 1. $\mathbf{B}_{\cdot, i}$ denotes the $i^{\text{th}}$ column of $\mathbf{B}$, and $x$ a realization of $X$ regarding a user. Besides $\mathbf{B}$, the vector of *conditional* variances $\vec{\sigma}^2 := (\sigma_1^2, ..., \sigma_m^2)$ are further model parameters: $\text{var}(X_i | X_{\mathcal{I} \setminus \{i\}} = x_{\mathcal{I} \setminus \{i\}}) = \sigma_i^2$. The fact that the covariance matrix $\Sigma$ is symmetric imposes the constraint $\sigma_i^2 \beta_{i,j} = \sigma_j^2 \beta_{j,i}$ on the auto-normal parameterization [7]. Moreover, the positive definiteness of $\Sigma$ gives rise to an additional constraint, which in general can only be verified if the numerical values of the parameters are known [7]. For computational efficiency, we will not explicitly enforce either one of these constraints in this paper.

### 2.2 Parameter Fitting

Besag's *pseudo-likelihood* yields asymptotically consistent estimates when the auto-normal parameterization is used [7].[3] The log pseudo-likelihood is defined as the sum of the conditional log likelihoods of the items: $L^{(\text{pseudo})}(\mathbf{X} | \mathbf{B}, \vec{\sigma}^2) = \sum_{i \in \mathcal{I}} L(\mathbf{X}_{\cdot, i} | \mathbf{X}_{\cdot, \mathcal{I} \setminus \{i\}}; \mathbf{B}_{\cdot, i}, \sigma_i^2)$, where $\mathbf{X}$ is the given user-item-interaction data-matrix $\mathbf{X} \in \mathbb{R}^{n \times m}$ regarding $n$ users and $m$ items. While this approach allows for a real-valued data-matrix $\mathbf{X}$ (e.g., the duration that a user listened to a song), in our experiments in Section 5, following the experimental set-up in [33], we use a *binary* matrix $\mathbf{X}$, where 1 indicates an observed user-item interaction (e.g., a user listened to a song). $\mathbf{X}_{\cdot, i}$ denotes column $i$ of matrix $\mathbf{X}$, while column $i$ is dropped in $\mathbf{X}_{\cdot, \mathcal{I} \setminus \{i\}}$. Note that we assume i.i.d. data. Substituting the Gaussian density function for each (univariate) conditional likelihood, results in $L^{(\text{pseudo})}(\mathbf{X} | \mathbf{B}, \vec{\sigma}^2) = -\sum_{i \in \mathcal{I}} \left\{ \frac{1}{2\sigma_i^2} ||\mathbf{X}_{\cdot, i} - \mathbf{X}\mathbf{B}_{\cdot, i}||_2^2 + \frac{1}{2} \log 2\pi\sigma_i^2 \right\}$. If the symmetry-constraint (cf. $\sigma_i^2 \beta_{i,j} = \sigma_j^2 \beta_{j,i}$ in previous section) is dropped, the parameters $\hat{\mathbf{B}}$ that maximize this pseudo-likelihood also maximize the decoupled pseudo-likelihood

$$L^{(\text{decoupled pseudo})}(\mathbf{X} | \mathbf{B}) = -\sum_{i \in \mathcal{I}} ||\mathbf{X}_{\cdot, i} - \mathbf{X}\mathbf{B}_{\cdot, i}||_2^2 = -||\mathbf{X} - \mathbf{X}\mathbf{B}||_F^2, \tag{2}$$

where $||\cdot||_F$ denotes the Frobenius norm of a matrix. Note that any weighting scheme $w_i > 0$, including $w_i = 1/(2\sigma_i^2)$ and $w_i = 1$, in $\sum_{i\in\mathcal{I}} w_i ||\mathbf{X}_{\cdot,i} - \mathbf{XB}_{\cdot,i}||_2^2$ results in the same optimum $\hat{\mathbf{B}}$. This is obvious from the fact that this sum is optimized by optimizing each column $\mathbf{B}_{\cdot,i}$ independently of the other columns, as they are decoupled in the absence of the symmetry constraint. Note that, unlike the pseudo-likelihood, Eq. 2 becomes independent of the (unknown) *conditional* variances $\sigma_i^2$.

### 2.2.1 Complete Graph

The result for the complete graph is useful for the next section. Starting from Eq. 2, we add L2-norm regularization with hyper-parameter $\lambda > 0$:

$$\hat{\mathbf{B}} = \arg\min_B ||\mathbf{X} - \mathbf{XB}||_F^2 + \lambda \cdot ||\mathbf{B}||_F^2 \quad \text{where} \quad \text{diag}(\mathbf{B}) = 0 \tag{3}$$

where we explicitly re-stated the zero-diagonal constraint, see Section 2.1. The method of Lagrangian multipliers immediately yields the closed-form solution (see derivation below):

$$\hat{\mathbf{B}} = \mathbf{I} - \hat{\mathbf{C}} \cdot \text{dMat}(1 \oslash \text{diag}(\hat{\mathbf{C}})) \quad \text{where} \quad \hat{\mathbf{C}} = \mathbf{S}_\lambda^{-1} \quad \text{and} \quad \mathbf{S}_\lambda = n^{-1}(\mathbf{X}^\top\mathbf{X} + \lambda \cdot \mathbf{I}), \tag{4}$$

where $\mathbf{I}$ denotes the identity matrix, $\text{dMat}(\cdot)$ a diagonal matrix, $\oslash$ the elementwise division, and $\text{diag}(\cdot)$ the diagonal of the estimated concentration matrix $\hat{\mathbf{C}}$, which is the inverse of the L2-norm regularized empirical covariance matrix $\mathbf{S}_\lambda$. Note that $\hat{\mathbf{B}}_{i,j} = -\hat{\mathbf{C}}_{i,j}/\hat{\mathbf{C}}_{j,j}$ for $i \neq j$.

**Derivation:** Eq. 3 can be solved via the method of Lagrangian multipliers: setting the derivative of the Lagrangian $||\mathbf{X} - \mathbf{XB}||_F^2 + \lambda \cdot ||\mathbf{B}||_F^2 + 2\gamma^\top \cdot \text{diag}(\mathbf{B})$ to zero, where $\gamma \in \mathbb{R}^m$ is the vector of Lagrangian multipliers regarding the equality constraint $\text{diag}(\mathbf{B}) = 0$, it follows after re-arranging terms: $\hat{\mathbf{B}} = (\mathbf{X}^\top\mathbf{X} + \lambda\cdot\mathbf{I})^{-1}\left(\mathbf{X}^\top\mathbf{X} - \text{dMat}(\gamma)\right) = n^{-1}\hat{\mathbf{C}}(n\hat{\mathbf{C}}^{-1} - \lambda\cdot\mathbf{I} - \text{dMat}(\gamma)) = \mathbf{I} - n^{-1}\hat{\mathbf{C}}\cdot\text{dMat}(\gamma + \lambda)$, where $\gamma$ is determined by the constraint $0 = \text{diag}(\hat{\mathbf{B}}) = \text{diag}(\mathbf{I}) - n^{-1}\text{diag}(\hat{\mathbf{C}}) \odot (\gamma + \lambda)$, where $\odot$ denotes the elementwise product. Hence, $\gamma + \lambda = n \oslash \text{diag}(\hat{\mathbf{C}})$. $\square$

### 2.2.2 Subgraphs

The result from the previous section carries immediately over to certain subgraphs:

**Corollary:** Let $\mathcal{D} \subseteq \mathcal{I}$ be a subset of nodes that forms a fully connected subgraph in $G$. Let $\mathcal{C}$ be the Markov blanket of $\mathcal{D}$ in graph $G$ such that each $j \in \mathcal{C}$ is connected to each $i \in \mathcal{D}$. Then the non-zero parameter-estimates in the columns $i \in \mathcal{D}$ of $\hat{\mathbf{B}}$ based on the pseudo-likelihood are asymptotically consistent, and given by $\hat{\mathbf{B}}_{j,i} = -\hat{\mathbf{C}}_{j,i}/\hat{\mathbf{C}}_{i,i}$ for all $i \in \mathcal{D}$ and $j \in \mathcal{C} \cup \mathcal{D} \setminus \{i\}$, where the submatrix of matrix $\hat{\mathbf{C}} \in \mathbb{R}^{|\mathcal{I}|\times|\mathcal{I}|}$ regarding the nodes $\mathcal{C} \cup \mathcal{D}$ is determined by the inverse of the submatrix of the empirical covariance matrix:[4] $\hat{\mathbf{C}}[\mathcal{C} \cup \mathcal{D}; \mathcal{C} \cup \mathcal{D}] = \mathbf{S}_\lambda[\mathcal{C} \cup \mathcal{D}; \mathcal{C} \cup \mathcal{D}]^{-1}$.

**Proof:** This follows trivially when considering the nodes in $\mathcal{D}$ as the so-called *dependents* and the nodes in $\mathcal{C}$ as the *conditioners* in the *coding technique* used in [7]. The estimate in Eq. 4 for the complete graph carries over to the nodes $\mathcal{D}$, as each $i \in \mathcal{D}$ is connected to all $j \in \mathcal{C} \cup \mathcal{D}$, and $\mathcal{D}$ given the conditioners $\mathcal{C}$ is independent of all remaining nodes in graph $G$. $\square$

## 3 Sparse Approximation

In collaborative-filtering problems with a large number of items, the graph $G$ can be expected to be (approximately) sparse, where related items form densely connected subgraphs, while items in different subgraphs are only sparsely connected. An absent edge in graph $G$ is equivalent to a zero entry in the concentration matrix $\mathbf{C}$ [31, 36] and in the matrix of regression coefficients $\mathbf{B}$ (see Eq. 4).

In our approach, the goal is to trade accuracy for training-time, rather than to learn the 'true' graph $G$ and the most accurate parameters at any computational cost. This is important in practical applications, where recommender systems have to be re-trained regularly under time-constraints as to ingest the most recent data. To this end, we use model-sparsity as a means for speeding up the training (rather than for improving accuracy), as it reduces the number of parameters that need to be learned. The Corollary outlined above can be used for an approximation where a large number of small *submatrices*

of the concentration matrix $\hat{\mathbf{C}}$ has to be inverted, each regarding a set of related items $\mathcal{D}$ conditioned on their Markov blanket $\mathcal{C}$. This can be computationally much more efficient (1) compared to inverting the entire concentration matrix $\hat{\mathbf{C}}$ at once (like in Eq. 4, which can be computationally expensive if the number of items is large), or (2) compared to regressing *each individual node* against its neighbors as is commonly done in the literature (e.g., [7, 21, 36, 38]).

Our approximation is comprised of two parts: first, the empirical covariance matrix is thresholded (see next section), resulting in a sparsity pattern that has to be sufficiently sparse as to enable computationally efficient estimation of the (approximate) regression coefficients in $\mathbf{B}$ in the second part (see Section 3.2). An implementation of the algorithm, and the used values of the hyper-parameters, are publicly available at `https://github.com/hasteck/MRF_NeurIPS_2019`.

## 3.1 Approximate Graph Structure

Numerous approaches for learning the sparse graph structure (and parameters) have been proposed in recent years, e.g., [36, 59, 15, 2, 60, 44, 45, 63, 55, 24, 25, 52, 56, 51]. Interestingly, simply applying a threshold to the empirical covariance matrix (in absolute value) [10, 9, 17, 57, 35, 48, 14, 61, 13] can recover the same sparsity pattern as the graphical lasso does [59, 15, 2] under certain assumptions, regarding the connected components [57, 35], as well as the edges [48, 14, 61, 13] in the graph. While it may be computationally expensive to verify that the underlying assumptions are met by a given empirical covariance matrix, the rule of thumb given in [48] is that the assumptions can be expected to hold if the resulting matrix is 'very' sparse. For computational efficiency, we hence apply a threshold to $\mathbf{S}_\lambda$ as to obtain a sufficiently sparse matrix $\mathbf{A} \in \mathbb{R}^{|\mathcal{I}| \times |\mathcal{I}|}$ reflecting the sparsity pattern.

Additionally, we apply an upper limit on the number of non-zero entries per column in $\mathbf{A}$ (retaining the entries with the largest values in $\mathbf{S}_\lambda$), as to bound the maximal training-cost of each iteration (see second to last paragraph in the in the next section). We allow at most 1,000 non-zero entries per column in our experiments in Section 5, based on the trade-off between training time and prediction accuracy: a smaller value tends to reduce the training-time, but it might also degrade the prediction accuracy of the learned sparse model. In Table 2, this threshold actually affects only about 2% of the items when using the sparsity level of 0.5%, while it has no effect at the sparsity level of 0.1%. Apart from that, allowing an item to have up to 1,000 similar items (e.g., songs in the *MSD* data) seems a reasonably large number in practice.

## 3.2 Approximate Parameter-Estimates

In this section, we outline our novel approach for approximate parameter-estimation given the sparsity pattern $\mathbf{A}$ from the previous section. In this approach, the trade-off between approximation-accuracy and training-time is controlled by the value of the hyper-parameter $r \in [0, 1]$ used in step 2 below.

Given the sparsity pattern $\mathbf{A}$, we first create a list $\mathcal{L}$ of all items $i \in \mathcal{I}$, sorted in descending order by each item's number of neighbors in $\mathbf{A}$, i.e., number of non-zero entries in column $i$ (ties may be broken according to the items' popularities). Our iterative approach is based on this list $\mathcal{L}$, which gets modified until it is empty, which marks the end of the iterations. We also use a set $\mathcal{S}$, initialized to be empty. Each iteration is comprised of the following four steps:

**Step 1:** We take the first element from list $\mathcal{L}$, say item $i$, and insert it into set $\mathcal{S}$. Then we determine its neighbors $\mathcal{N}^{(i)}$ based on the $i^{\text{th}}$ column of the sparsity pattern in matrix $\mathbf{A}$.

**Step 2:** We now split the set $\mathcal{N}^{(i)} \cup \{i\}$ into two disjoint sets such that set $\mathcal{D}^{(i)}$ contains node $i$ as well as the $m^{(i)} = \text{round}(r \cdot |\mathcal{N}^{(i)}|)$ nodes that have the largest empirical covariances with node $i$ (in absolute value), where $r \in [0, 1]$ is the chosen hyper-parameter. The second set is $\mathcal{C}^{(i)} := (\mathcal{N}^{(i)} \cup \{i\}) \setminus \mathcal{D}^{(i)}$. We now make the key assumption of this approach (and do not verify it as to save computation time), namely that $\mathcal{C}^{(i)}$ is a Markov blanket of $\mathcal{D}^{(i)}$ in the sparse graph $G$. Obviously, this is a strong assumption, and cannot be expected to hold in general. It may not be unreasonable, however, to expect this to be an approximation in the sense that $\mathcal{C}^{(i)}$ contains many nodes of the (actual) Markov blanket of $\mathcal{D}^{(i)}$ in graph $G$ for two reasons: (1) if $m^{(i)} = 0$, then $\mathcal{C}^{(i)} = \mathcal{N}^{(i)}$ is indeed the Markov blanket of $\mathcal{D}^{(i)} = \{i\}$; (2) given that we chose $\mathcal{D}^{(i)}$ to contain the variables with the largest covariances to node $i$, their Markov blankets likely have many nodes in common with the Markov blanket of node $i$. As we increase the value of $m^{(i)} \leq |\mathcal{N}^{(i)}|$, the

approximation-accuracy obviously deteriorates (except for the case that $\mathcal{N}^{(i)} \cup \{i\}$ is a connected component in graph $G$). For these reasons, the value of $m^{(i)}$ (which is controlled by the chosen value of $r$) allows one to control the trade-off between approximation accuracy and computational cost.

**Step 3:** Given set $\mathcal{D}^{(i)}$ and its (assumed) Markov blanket $\mathcal{C}^{(i)}$, we now assume that these nodes are connected as required by the Corollary above. Note that this may assume additional edges to be present, resulting in additional regression parameters that need to be estimated. Obviously, this is a further approximation. However, the decrease in statistical efficiency can be expected to be rather small in the typical setting of collaborative filtering, where the number of data points (i.e., users) usually is much larger than the number of nodes (i.e., items) in the (typically small) subset $\mathcal{D}^{(i)} \cup \mathcal{C}^{(i)}$. We now can apply the Corollary in Section 2.2.2, and obtain the estimates for all the columns $j \in \mathcal{D}^{(i)}$ in matrix $\hat{\mathbf{B}}$ at once. This is the key to the computational efficiency of this approach: for about the same computational cost as estimating the single column $i$ in $\hat{\mathbf{B}}$, we now obtain the (approximate) estimates for $1 + m^{(i)}$ columns (see Section 2.2.2 for details).

**Step 4:** Finally, we remove all the $1 + m^{(i)}$ items in $\mathcal{D}^{(i)}$ from the sorted list $\mathcal{L}$, and go to step 1 unless $\mathcal{L}$ is empty. Obviously, as we increase the value of $r$ (and hence $m^{(i)}$), the size of list $\mathcal{L}$ decreases by a larger number in each iteration, eventually requiring fewer iterations, which reduces the training-time. If we choose $m^{(i)} = 0$, then $\mathcal{D}^{(i)} = \{i\}$, and there is no computational speed-up (and also no approximation) compared to the baseline of solving one regression problem per column in $\hat{\mathbf{B}}$ w.r.t. the pseudo-likelihood.

**Upon completion of the iterations,** we have estimates for all columns of $\hat{\mathbf{B}}$. In fact, for many entries $(j, i)$, there may be multiple estimates; for instance if node $i \in \mathcal{D}^{(k)}$ and node $j \in \mathcal{D}^{(k)} \cup \mathcal{C}^{(k)}$ for several different nodes $k \in \mathcal{S}$. As to aggregate possibly multiple estimates for entry $\hat{\mathbf{B}}_{j,i}$ into a single value, we simply use their average in our experiments.

The computational complexity of this iterative scheme can be controlled by the sparsity level chosen in Section 3.1, as well as the chosen value $r \in [0, 1]$, which determines the values $m^{(i)}$ (see step 2). When using the Coppersmith-Winograd algorithm for matrix inversion, it is given by $\mathcal{O}(\sum_{i \in \mathcal{S}} (1 + |\mathcal{N}^{(i)}|)^{2.376})$, where the size of $\mathcal{S}$ depends on the chosen values $r$. Note that the sum may be dominated by the largest value $|\mathcal{N}^{(i)}|$, which motivated us to cap this value in Section 3.1. Note that set $\mathcal{S}$ can be computed in linear time in $|\mathcal{I}|$ by iterating through steps 1, 2, and 4 (skipping step 3). Once $\mathcal{S}$ is determined, the computation of step 3 for different $i \in \mathcal{S}$ is embarrassingly parallel. In comparison, in the standard approach of separately regressing each item $i$ against its neighbors $\mathcal{N}^{(i)}$, we have $\mathcal{O}(\sum_{i \in \mathcal{I}} |\mathcal{N}^{(i)}|^{2.376})$, i.e., the sum here extends over all $i \in \mathcal{I}$ (instead of subset $\mathcal{S} \subseteq \mathcal{I}$ only). In the other extreme, inverting the entire covariance matrix incurs the cost $\mathcal{O}(|\mathcal{I}|^{2.376})$.

Lacking an analytical error-bound, the accuracy of this approximation may be assessed empirically, by simply learning $\hat{\mathbf{B}}$ under different choices regarding the sparsity level (see Section 3.1) and the value $r$ (see step 2 above). Given that recommender systems are re-trained frequently as to ingest the most recent data, only these regular updates require efficient computations in practice. Additional models with higher accuracy (and increased training-time) may be learned occasionally as to assess the accuracy of the models that get trained regularly.

# 4 Related Work

We discuss the connections to various related approaches in this section.

Several **non-Gaussian distributions** are also covered by Besag's auto-models [5, 6], including the logistic auto-model for binary data. Binary data were also considered in [2, 42]. While we rely on the Gaussian distribution for computational efficiency, note that, regarding model-fit, Eq. 4 and the Corollary provide the best least-squares fit of a linear model for any distribution of $X$, as Eq. 4 is the solution of Eq. 3. The empirical results in Section 5 corroborate that this is an effective trade-off between accuracy and training-time.

**Sparse Inverse Covariance Estimation** has seen tremendous progress beyond the graphical lasso [59, 15, 2]. A main focus was computationally efficient optimization of the full likelihood [24, 25, 52, 44, 51, 55, 45], often in the regime where $n < m$ (e.g., [24, 52, 51]), or the regime of small data (e.g., [63, 56]). Node-wise regression was considered for structure-learning in [36] and

for parameter-learning in [60], which is along the lines of Besag's pseudo-likelihood [7, 8]. The pseudo-likelihood was generalized in [34]. Our paper focuses on a different regime, with large $n, m$ and $n > m$, as typical for collaborative filtering. In Section 3.2, we outlined a novel kind of sparse approximation, using set-wise rather than the node-wise regression, which is commonly used in the literature.

**Dependency Networks** [21] also regress each node against its neighbors. As this may result in inconsistencies when learning from finite data, a kind of Gibbs sampling is used as to obtain a consistent joint distribution. This increases the computational cost. Given that collaborative filtering typically operates in the regime of large $n, m$ and $n > m$, we rely on the asymptotic consistency of Besag's pseudo-likelihood for computational efficiency.

In $\textsc{Slim}$ [38], the objective is similar to Eq. 3, but is comprised of two additional terms: (1) sparsity-promoting L1-norm regularization and (2) a non-negativity constraint on the learned regression parameters. As we can see in Table 1, this not only reduces accuracy but also increases training-time, compared to $\hat{\mathbf{B}}^{(\mathrm{dense})}$. In [38], also the variant fsSLIM was proposed, where first the sparsity pattern was determined via a k-nearest-neighbor approach, and then a separate regression problem was solved for each node. This node-wise regression is a special case (for $r = 0$) of our set-based approximation outlined in Section 3.2. The variants proposed in [46, 32] drop the constraint of a zero diagonal for computational efficiency, which however is an essential property of Besag's auto-models [7]. The logistic loss is used in [46], which requires one to solve a separate logistic regression problem for each node, which is computationally expensive.

**Autoencoders and Deep Learning** have led to many improvements in collaborative filtering [23, 22, 33, 47, 62, 58, 20, 11]. In the pseudo-likelihood of the MRF in Eq. 3, the objective is to reproduce $\mathbf{X}$ from $\mathbf{X}$ (using $\mathbf{B}$), like in an autoencoder, cf. also our short paper [50]. However, there is no encoder, decoder or hidden layer in the MRF in Eq. 3. The learned $\mathbf{B}$ is typically of full rank, and the constraint $\mathrm{diag}(\mathbf{B}) = 0$ is essential for generalizing to unseen data. The empirical evidence in Section 5 corroborates that this is a viable alternative to using low-dimensional embeddings, as in typical autoencoders, as to generalize to unseen data. In fact, recent work on deep collaborative filtering combines low-rank and full-rank models for improved recommendation accuracy [11]. Moreover, recent progress in deep learning has also led to full-rank models, like invertible deep networks [27], as well as flow-based generative models [16, 30, 40, 12, 29]. Adapting these approaches to collaborative filtering appears to be promising future work in light of the experimental results obtained by the full-rank shallow model in this paper.

**Neighborhood Approaches** are typically based on a heuristic item-item (or user-user) similarity matrix (e.g. cosine similarity), e.g., [1, 53, 54] and references therein. Our approach yields three key differences to cosine-similarity and the like: (1) a principled way of learning/optimizing the similarity matrix $\hat{\mathbf{B}}$ from data; (2) Eq. 4 shows that the conceptually correct similarity matrix is not based on (a re-scaled version of) the covariance matrix, but on its inverse; (3) the similarity matrix is asymmetric (cf. Eq. 4) rather than symmetric.

## 5  Experiments

In our experiments, we empirically evaluate the closed-form (dense) solution $\hat{\mathbf{B}}^{(\mathrm{dense})}$ (see Eq. 4) as well as the sparse approximation outlined in Section 3. We follow the experimental set-up in [33] and use their publicly available code for reproducibility.[5] Three well-known data sets were used in the experiments in [33]: MovieLens 20 Million (***ML-20M***) [19], Netflix Prize (***Netflix***) [3], and the Million Song Data (***MSD***) [4]. They were pre-processed and filtered for items and users with a certain activity level in [33], resulting in the data-set sizes shown in Table 1. We use all the approaches evaluated on these three data-sets in [33] as baselines:

- Sparse Linear Method ($\textsc{Slim}$) [38] as discussed in Section 4.
- Weighted Matrix Factorization ($\textsc{wmf}$) [26, 39]: A linear model with a latent representation of users and items. Variants like NSVD [41] or FISM [28] obtained very similar accuracies.
- Collaborative Denoising Autoencoder ($\textsc{cdae}$) [58]: nonlinear model with 1 hidden layer.

- Denoising Autoencoder (**MULT-DAE**) and Variational Autoencoder (**MULT-VAE**) [33]: deep nonlinear models, trained w.r.t. the multinomial likelihood. Three hidden layers were found to obtain the best accuracy on these data-sets, see Section 4.3 in [33]. Note that rather shallow architectures are commonly found to obtain the highest accuracy in collaborative-filtering (which is different from other application areas of deep learning, like image classification, where deeper architectures often achieve higher accuracy).

We do not compare to Neural Collaborative Filtering (NCF), its extension NeuCF [20] and to Bayesian Personalized Ranking (BPR) [43], as their accuracies were found to be below par on the three data-sets *ML-20M*, *Netflix*, and *MSD* in [33]. NCF and NeuCF [20] was competitive only on unrealistically small data-sets in [33].

We follow the evaluation protocol used in [33],[5] which is based on *strong generalization*, i.e., the training, validation and test sets are disjoint in terms of the users. Normalized Discounted Cumulative Gain (nDCG@100) and Recall (@20 and @50) served as ranking metrics for evaluation in [33]. For further details of the experimental set-up, the reader is referred to [33].

Note that the training-data matrix $\mathbf{X}$ here is *binary*, where 1 indicates an observed user-item interaction. This obviously violates our assumption of a Gaussian distribution, which we made for reasons of computational efficiency. In this case, our approach yields the best least-squares fit of a linear model, as discussed in Section 4. The empirical results corroborate that this is a viable trade-off between accuracy and training-time, as discussed in the following.

**Closed-Form Dense Solution:** Table 1 summarizes the experimental results across the three data sets. It shows that the closed-form solution $\hat{\mathbf{B}}^{(\mathrm{dense})}$ (see Eq. 4) obtains nDCG@100 that is about 1% lower on *ML-20M*, about 3% better on *Netflix*, and a remarkable 24% better on *MSD* than the best competing model, MULT-VAE.

It is an interesting question as to why this simple full-rank model outperforms the deep nonlinear MULT-VAE by such a large margin on the *MSD* data. We suspect that the hourglass architecture of MULT-VAE (where the smallest hidden layer has 200 dimensions in [33]) severely restricts the information that can flow between the 41,140-dimensional input and output layers (regarding the 41,140 items in *MSD* data), so that many relevant dependencies between items may get lost. For instance, compared to the full-rank model, MULT-VAE recommends long-tail items considerably less frequently among the top-N items, on average across all test users in the *MSD* data, see also [50]. As the *MSD* data contain about twice as many items as the other two data sets, this would

Table 1: The closed-form dense solution $\hat{\mathbf{B}}^{(\mathrm{dense})}$ (see Eq. 4) obtains competitive ranking-accuracy while requiring only a small fraction of the training time, compared to the various models empirically evaluated in [33]. The standard errors of the ranking-metrics are about 0.002, 0.001, and 0.001 on *ML-20M*, *Netflix*, and *MSD* data [33], respectively.

| models | *ML-20M* nDCG @100 | Recall @20 | Recall @50 | *Netflix* nDCG @100 | Recall @20 | Recall @50 | *MSD* nDCG @100 | Recall @20 | Recall @50 |
|---|---|---|---|---|---|---|---|---|---|
| $\hat{\mathbf{B}}^{(\mathrm{dense})}$ | 0.423 | 0.392 | 0.522 | 0.397 | 0.364 | 0.448 | 0.391 | 0.334 | 0.430 |
| reproduced from [33]: | | | | | | | | | |
| MULT-VAE | 0.426 | 0.395 | 0.537 | 0.386 | 0.351 | 0.444 | 0.316 | 0.266 | 0.364 |
| MULT-DAE | 0.419 | 0.387 | 0.524 | 0.380 | 0.344 | 0.438 | 0.313 | 0.266 | 0.363 |
| CDAE | 0.418 | 0.391 | 0.523 | 0.376 | 0.343 | 0.428 | 0.237 | 0.188 | 0.283 |
| SLIM | 0.401 | 0.370 | 0.495 | 0.379 | 0.347 | 0.428 | –did not finish in [33]– | | |
| WMF | 0.386 | 0.360 | 0.498 | 0.351 | 0.316 | 0.404 | 0.257 | 0.211 | 0.312 |
| **training times** | | | | | | | | | |
| $\hat{\mathbf{B}}^{(\mathrm{dense})}$ | 2 min 0 sec | | | 1 min 30 sec | | | 15 min 45 sec | | |
| MULT-VAE | 28 min 10 sec | | | 1 hour 26 min | | | 4 hours 30 min | | |
| **data-set properties** | 136,677 users 20,108 movies 10 million interactions | | | 463,435 users 17,769 movies 57 million interactions | | | 571,355 users 41,140 songs 34 million interactions | | |

also explain why the difference in ranking accuracy between MULT-VAE and the proposed full-rank model is the largest on the *MSD* data. While the ranking accuracy of MULT-VAE may be improved by considerably increasing the number of dimensions, note that this would prolong the training time at least linearly, which is already 4 hours 30 minutes for MULT-VAE on *MSD* data (see Table 1). Apart from that, as a simple sanity check, once the full-rank matrix $\hat{\mathbf{B}}^{(\text{dense})}$ was learned, we applied a low-rank approximation (SVD), and found that even 3,000 dimensions resulted in about a 10% drop in nDCG@100 on *MSD* data. This motivated us to pursue sparse full-rank rather than dense low-rank approximations in this paper, which is naturally facilitated by MRFs.

Besides the differences in accuracy, Table 1 also shows that the training-times of MULT-VAE are more than ten times larger than the few minutes required to learn $\hat{\mathbf{B}}^{(\text{dense})}$ on all three data-sets. The reasons are that MULT-VAE is trained on the user-item data-matrix $\mathbf{X}$ and uses stochastic gradient descent to optimize ELBO (which involves several expensive computations in each step)–in contrast, the proposed MRF uses a closed-form solution, and is trained on the item-item data-matrix (note that #items $\ll$ #users in our experiments). Also note that the training-times of MULT-VAE reported in Table 1 are optimistic, as they are based on only 50 iterations, where the training of MULT-VAE may not have fully converged yet (the reported accuracies of MULT-VAE are based on 200 iterations). These times were obtained on an AWS instance with 64 GB memory and 16 vCPUs for learning $\hat{\mathbf{B}}^{(\text{dense})}$, and with a GPU for training MULT-VAE (which was about five times faster than training MULT-VAE on 16 vCPUs).

**Sparse Approximation:** Given the short training-times of the closed-form solution on these three data-sets, we demonstrate the speed-up obtained by the sparse approximation (see Section 3) on the *MSD* data, where the training of the closed-form solution took the longest: Table 2 shows that the training-time can be reduced from about 16 minutes for the closed-form solution to under a minute with only a relatively small loss in accuracy: while the loss in accuracy is statistically significant (standard error is about 0.001), it is still very small compared to the difference to MULT-VAE, the most accurate competing model in Table 1.

We can also see in Table 2 that different trade-offs between accuracy and training-time can be obtained by using a sparser model and/or a larger hyper-parameter $r$ (which increases the sizes of the *subsets* of items in step 2 in Section 3.2): first, the special case $r = 0$ corresponds to regressing *each individual item* (instead of a subset of items) against its neighbors in the MRF, which is commonly done in the literature (e.g., [7, 21, 36, 38]). Table 2 illustrates that this can be computationally more expensive than inverting the entire covariance matrix at once (cf. MRF with sparsity 0.5% and $r = 0$ vs. the dense solution). Second, comparing the MRF with sparsity 0.5% and $r = 0.5$ vs. the model with sparsity 0.1% and $r = 0$, we can see that the former obtains a better Recall@50 than the latter does, and also requires less training time. This illustrates that it can be beneficial to learn a denser model (0.5% vs. 0.1% sparsity here) but with a larger value $r$ (0.5 vs. 0 here). Note that optimizing Recall@50 (vs. @20) is important in applications where a large number of items has to

Table 2: Sparse Approximation (see Section 3), on *MSD* data (standard error $\approx 0.001$): ranking-accuracy can be traded for training-time, controlled by the sparsity-level and the parameter $r \in [0, 1]$ (defined in Section 3). For comparison, also the closed-form solution $\hat{\mathbf{B}}^{(\text{dense})}$ and the best competing model, MULT-VAE, from Table 1 are shown.

| | nDCG@100 | Recall@20 | Recall@50 | Training Time |
|---|---|---|---|---|
| $\hat{\mathbf{B}}^{(\text{dense})}$ | 0.391 | 0.334 | 0.430 | 15 min 45 sec |
| 0.5% sparse approximation (see Section 3) | | | | |
| $r = 0$ | 0.390 | 0.333 | 0.427 | 21 min 12 sec |
| $r = 0.1$ | 0.387 | 0.331 | 0.424 | 3 min 27 sec |
| $r = 0.5$ | 0.385 | 0.330 | 0.424 | 2 min 1 sec |
| 0.1% sparse approximation (see Section 3) | | | | |
| $r = 0$ | 0.385 | 0.330 | 0.421 | 3 min 7 sec |
| $r = 0.1$ | 0.382 | 0.327 | 0.417 | 1 min 10 sec |
| $r = 0.5$ | 0.381 | 0.327 | 0.417 | 39 sec |
| MULT-VAE | 0.316 | 0.266 | 0.364 | 4 hours 30 min |

be recommended, like for instance on the homepages of video streaming services, where typically hundreds of videos are recommended. The proposed sparse approximation enables one to choose the optimal trade-off between training-time and ranking-accuracy for a given real-world application.

Also note that, at sparsity levels 0.5% and 0.1%, our sparse model contains about the same number of parameters as a dense matrix of size 41,140×200 and 41,140×40, respectively. In comparison, MULT-VAE in [33] is comprised of layers with dimensions 41,140 → 600 → 200 → 600 → 41,140 regarding the 41,140 items in the *MSD* data, i.e., it uses two matrices of size 41,140×600. Hence, our sparse approximation (1) has only a fraction of the parameters, (2) requires orders of magnitude less training time, and (3) still obtains about 20% better a ranking-accuracy than MULT-VAE in Table 2, the best competing model (see also Table 1).

**Popularity Bias:** The popularity bias in the model's predictions is very important for obtaining high recommendation accuracy, see also [49]. The different item-popularities affect the means and the covariances in the Gaussian MRF, and we used the standard procedure of centering the user-item matrix $\mathbf{X}$ (zero mean) and re-scaling the columns of $\mathbf{X}$ prior to training (once the training was completed, and when making predictions, we scaled the predicted values back to the original space, so that the predictions reflected the full popularity bias in the training data, which can be expected to be the same as the popularity bias in the test data due to the way the data were split). This is particularly important when learning the sparse model: theoretically, its sparsity pattern is determined by the *correlation* matrix (which quantifies the strength of statistical dependence between the nodes in the Gaussian MRF), while the values (after scaling them back to the original space) of the non-zero entries are determined by the *covariance* matrix. In practice, we divided each column $i$ in $\mathbf{X}$ by $s_i = \mathrm{std}_i^{\alpha}$, where $\mathrm{std}_i$ is the column's empirical standard deviation; the grid search regarding the exponent $\alpha \in \{0, 1/4, 1/2, 3/4, 1\}$ yielded the best accuracy for $\alpha = 3/4$ on *MSD* data (note that $\alpha = 1$ would result in the correlation matrix), which coincidentally is the same value as was used in word2vec [37] to remove the word-popularities in text-data as to learn word-similarities.

## Conclusions

Geared toward collaborative filtering, where typically the number of users $n$ (data points) and items $m$ (variables) are large and $n > m$, we presented a computationally efficient approximation to learning sparse Gaussian Markov Random Fields (MRF). The key idea is to solve a large number of regression problems, each regarding a *small subset* of items. The size of each subset can be controlled, and this enables one to trade accuracy for training-time. As special (and extreme) cases, it subsumes the approaches commonly considered in the literature, namely regressing *each item* (i.e., set of size one) against its neighbors in the MRF, as well as inverting the entire covariance matrix at once. Apart from that, the auto-normal parameterization of MRFs prevents self-similarity of items (i.e., zero-diagonal in the weight-matrix), which we found an effective alternative to using low-dimensional embeddings in autoencoders in our experiments, as to enable the learned model to generalize to unseen data. Requiring several orders of magnitude less training time, the proposed sparse approximation resulted in a model with fewer parameters than the competing models, while obtaining about 20% better ranking accuracy on the data-set with the largest number of items in our experiments.

### Appendix

Let $\hat{\mu}^{\top} \neq \mathbf{0}$ denote the row vector of the (empirical) column-means of the given user-item data-matrix $\mathbf{X}$. If we assume that matrix $\mathbf{B}$ fulfills the eigenvector-constraint $\hat{\mu}^{\top} \cdot \mathbf{B} = \hat{\mu}^{\top}$, then it holds that $(\mathbf{X} - \mathbf{1} \cdot \hat{\mu}^{\top}) - (\mathbf{X} - \mathbf{1} \cdot \hat{\mu}^{\top}) \cdot \mathbf{B} = \mathbf{X} - \mathbf{X} \cdot \mathbf{B}$ (where $\mathbf{1}$ denotes a column vector of ones in the outer product with $\hat{\mu}^{\top}$). In other words, the learned $\hat{\mathbf{B}}$ is invariant under centering the columns of the training data $\mathbf{X}$. When the constraint $\hat{\mu}^{\top} \cdot \mathbf{B} = \hat{\mu}^{\top}$ is added to the training objective in Eq. 3, the method of Lagrangian multipliers again yields the closed-form solution:

$$\hat{\mathbf{B}} = I - \left( I - \frac{\hat{\mathbf{C}}\hat{\mu}\hat{\mu}^{\top}}{\hat{\mu}^{\top}\hat{\mathbf{C}}\hat{\mu}} \right) \cdot \hat{\mathbf{C}} \cdot \mathrm{dMat}(\tilde{\gamma}),$$

where now $\tilde{\gamma} = 1 \oslash \mathrm{diag}((I - \frac{\hat{\mathbf{C}}\hat{\mu}\hat{\mu}^{\top}}{\hat{\mu}^{\top}\hat{\mathbf{C}}\hat{\mu}}) \cdot \hat{\mathbf{C}})$ for the zero diagonal of $\hat{\mathbf{B}}$. The difference to Eq. 4 is merely the additional factor $I - \frac{\hat{\mathbf{C}}\hat{\mu}\hat{\mu}^{\top}}{\hat{\mu}^{\top}\hat{\mathbf{C}}\hat{\mu}}$ due to the constraint $\hat{\mu}^{\top} \cdot \mathbf{B} = \hat{\mu}^{\top}$. In our experiments, however, we did not observe this to cause any significant effect regarding the ranking metrics.

## Acknowledgments

I am very grateful to Tony Jebara for his encouragement, and to Dawen Liang for providing the code for the experimental setup of all three data-sets.

## Footnotes

[1] In fact, it is possible to drop this common assumption in certain cases, see Appendix.

[2] We use 'item', 'node' and 'random variable' interchangeably in this paper.

[3] Interestingly, despite its similarity to Eq. 1, the parameterization $X_i = \sum_{j \in \mathcal{I} \setminus \{i\}} \beta_{i,j} X_j + \epsilon_i$, where $\epsilon_i$ is independent Gaussian noise with zero mean and variance $\sigma_i^2$, does not lead to consistent estimates [7].

[4]When used as indices regarding a (sub-)matrix, a set of indices is used as if it were a list of sorted indices.

[5]The code regarding *ML-20M* in [33] is publicly available at `https://github.com/dawenl/vae_cf`, and can be modified for the other two data-sets as described in [33].

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
