[Reviews · NeurIPS 2019]

Reviewer 1



This paper proposes a sparse Gaussian MRF approximation for collaborative filtering applications. The proposed approach builds on a previous approach known as Besag’s pseudo-likelihood method for estimating a MRF under the auto-normal parameterization. The proposed sparse approximation model, and learning algorithm, appear to be novel. The learning algorithm is logically complex, although described reasonably well. However, given the complexity of the learning algorithm, it would be helpful for the authors to provide a high-level summary of the learning algorithm, perhaps via pseudocode. The experimental results shown in Table 1 and 2 are reasonably convincing, particularly regarding the training time speeds compared to the best performing baseline model. However, the authors should describe exactly why their proposed approach is orders of magnitude faster than the MULT-VAE baseline, perhaps by discussing the computational complexity of training MULT-VAE compared to the proposed approach. Additionally, it would be helpful for the authors to provide some explanation of why their proposed sparse MRF, which is a full-rank shallow model, is able to match or significantly exceed the performance of MULT-VAE, which is a deep nonlinear model. Furthermore, is the comparison with the MULT-VAE model with 3 hidden layers fair, or is it possible to further improve the predictive quality of MULT-VAE by adding additional hidden layers? Given the impressive training time performance and predictive quality, it seems likely that the proposed sparse MRF model could form the basis for future work that builds on this approach.

Reviewer 2



[UPDATE] Given the authors' response (which I find instructive and promising for the final version), I raise my score to a clear accept. The paper presents a novel method for recommendation with collaborative filtering based on Markov Random Fields (MRF). Starting from a general approach that regresses the full graph of items, the paper shows that a valid approximation can be obtained by proceeding with subgraphs that represent Markov blankets of an initial set of items. This approach yields significant computing gains, while yielding better recommendation performance compared to the state-of-the-art represented here by variational auto-encoders. ** Originality ** The paper presents original work on the topic with a new approach which generalises well to various datasets. As a general comment, I am wondering whether taking into account the popularity bias makes sense in the approach and if the authors thought about it. ** Quality ** I find this work to be of good quality overall. The claims are well supported by theoretical analysis. The experimental results are well documented and use well-known and appropriate datasets. Since the paper presents a new algorithm, I hope that the code will be submitted if the paper is accepted. It would have been nice to see the code during review, although the description of the algorithm in section 3.2 is clear and detailed. ** Significance ** The results are convincing in terms of performance and computing gains. There are benefits both in terms of recommendation performance and computing time. The state-of-the-art used as a benchmark is valid (variational auto encoders). The choice if 1,000 as a parameter (line 141) seems a bit arbitrary. It might be useful to explain how this parameter was chosen and if it impacts the results significantly. ** Clarity ** The method is presented in a clear and concise way. The authors acknowledge the fact that there is no proof of convergence, but illustrate the method in details on several reference datasets. The paper is well organised and clearly written. For the sake of clarity, it might be worth explaining how X is built early in section 2.2 as opposed to mentioning it only in section 5. Typo in reference [130] in the list of authors ("and" appears twice)

Reviewer 3



The paper is interesting and it is well-written. The methodology, algorithms, and results are easy to read and clear. My only concern is that this paper lack a solid theory. I believe that rather than yet another new algorithm that seems to work well, the community is mature enough for deeper theoretical results which are absent from this paper.

[Author Response · NeurIPS 2019]

We would like to thank the reviewers for their detailed reviews and their suggestions / questions, which will help to
further improve the clarity of this paper. In the following we will try to address the main questions.

**1. Three questions regarding MULT-VAE [33], one of the models we used as baseline (Reviewer 1):**

(a) In paper [33], where MULT-VAE was proposed, it was empirically found that **MULT-VAE with 3 hidden layers**
obtained the best accuracy on these data-sets, outperforming architectures with a larger (as well as a smaller) number of
hidden layers (see Section 4.3 in [33]). Note that this finding is consistent with the literature on collaborative filtering
by deep models (and different from other application areas of deep learning, where deeper is typically better). It hence
is fair to compare to MULT-VAE with 3 hidden layers.

(b) **Why the proposed full-rank model outperforms the deep non-linear MULT-VAE by a large margin on the**
*MSD* **data** is an excellent question. We are not aware of a definite answer in the literature, and hope that this paper may
spark more work in this area. Empirically, the number of long-tail items that get recommended in the top-N items (on
average across all test users in the *MSD* data) turns out to be considerably lower for MULT-VAE than it is for the full-rank
model. We suspect that the hourglass architecture of MULT-VAE (where the smallest hidden layer has 200 dimensions in
[33]) severely restricts the information that can flow between the 41,140-dimensional input and output layers (regarding
the 41,140 items in *MSD* data), so that many relevant dependencies between items may get lost (especially involving
long-tail items). Considerably increasing the number of dimensions may improve accuracy–however, the training time
would increase at least linearly, and it is already 4 hours 30 minutes for MULT-VAE on *MSD* data (see Table 2). As a
simple sanity check, once the full-rank $\hat{\mathbf{B}}^{(\text{dense})}$ was learned, we applied a low-rank approximation (SVD), and found
that even 3,000 dimensions resulted in about a 10% drop in nDCG@100 on *MSD* data. This motivated us to pursue
sparse full-rank rather than dense low-rank approximations.

(c) Before discussing as to **why the proposed training of the MRF is faster than learning MULT-VAE by an order**
**of magnitude or more**, note that MULT-VAE is not unusually time-consuming to train compared to various other
baseline models. For instance, in [33] (which proposed MULT-VAE and used SLIM as a baseline) it was stated that
parallelized grid search for the SLIM model took about two weeks on the *Netflix* data, and the *MSD* data-set was 'too
large for it to finish in a reasonable amount of time' [33]. At a high level, MULT-VAE is trained on the user-item
matrix $\mathbf{X}$ using stochastic gradient descent, which is time-consuming due to the large number of epochs required
until convergence (about 50 to 200 in [33]). Moreover, each gradient-step to optimize ELBO involves expensive
computations (log, exp, softmax, sampling, etc.). In contrast, the proposed MRF is trained on the item-item data-matrix
(note that #items $\ll$ #users in our experiments), and it uses a closed-form solution (instead of iterative gradient descent).

**2. High-level Summary and Pseudo-code (Reviewer 1):** While we described the high-level summary of the sparse
approximation in lines 107-114 and 325-329, we now realize that it may fit better at the beginning of Section 3. We will
also try to re-phrase this description to make it clearer. We omitted the pseudo-code in the paper due to space constraints,
but tried to write Section 4.2 in several individual steps as to resemble pseudo-code, but with the explanations included.
Based on the feedback, we will try to re-write this section more clearly as well.

**3. Code (Reviewer 2):** We plan to make our Python code available upon publication of this paper.

**4. Accounting for the Popularity Bias (Reviewer 2)** is indeed very important for obtaining high recommendation
accuracy. The different item-popularities affect the means and the covariances in the Gaussian MRF, and we used the
standard procedure of centering the user-item matrix $\mathbf{X}$ (see line 47 in our paper) and rescaling the columns of $\mathbf{X}$ prior
to training: see lines 314-321 for the exact approach, where $\alpha = 1$ results in the empirical correlation matrix (popularity
fully removed) and $\alpha = 0$ in the covariance matrix (popularity fully present). This is particularly important when
learning the sparse model: theoretically, its sparsity pattern is determined by the correlation matrix (which quantifies
the strength of statistical dependence between nodes in the Gaussian MRF), while the values of the non-zero entries are
determined by the covariance matrix (with the full popularities present, as the popularities of the items can be expected
to be the same in the test data and the training data, as these were obtained by randomly splitting the data in [33]).
Empirically, we found $\alpha = 3/4$ to result in slightly higher prediction accuracy than using the theoretically correct value
$\alpha = 1$ (i.e., correlation matrix) for determining the sparsity pattern, which coincidently is the same value as was used in
word2vec [Mikolov et al., NeurIPS 2013] to remove the word-popularities in text-data.

**5. We chose the threshold of at most 1,000 non-zero entries per column in the sparse approximation in Section**
**3.1 (Reviewer 2)** based on the tradeoff between training time and prediction accuracy: a smaller value tends to reduce
the training-time and computational complexity (see lines 138-141 and 190-192), but it might also degrade the prediction
accuracy of the learned sparse model. In Table 2, this threshold actually affects only a few dozen items in the model
with sparsity level 0.5%, while it has no effect at sparsity level 0.1% (where all items have fewer than 1,000 neighbors).
Apart from that, allowing an item (song) to have up to 1,000 similar songs in the *MSD* data seems a reasonably large
number based on our common sense.

[Meta-Review · NeurIPS 2019]

Reviewers were initially quite favorable with respect to this paper and your response lifted some remaining doubts (especially from Reviewer #1). I am happy to recommend acceptance, congratulations! I would recommend that you take the reviewer comments into account to prepare a camera-ready version. In particular, it seems to be important to incorporate some of the discussion in bullets 1 and 2 in your response (regarding Mult-VAE and the high-level summary or pseudocode).